# A Systematic Review and Meta-Analysis of Seasonal Influenza Vaccination of Health Workers

**DOI:** 10.3390/vaccines9101104

**Published:** 2021-09-29

**Authors:** Tingting Li, Xiaoling Qi, Qin Li, Wenge Tang, Kun Su, Mengmeng Jia, Weizhong Yang, Yu Xia, Yu Xiong, Li Qi, Luzhao Feng

**Affiliations:** 1Chongqing Municipal Center for Disease Control and Prevention, Chongqing 400016, China; Lee712ting@126.com (T.L.); liqin2006@163.com (Q.L.); twg@cqcdc.org (W.T.); sukun325@163.com (K.S.); xiayucq4@hotmail.com (Y.X.); Yuri811320@163.com (Y.X.); 2Affiliated Stomatological Hospital of Chongqing Medical University, Chongqing 400016, China; 501044@hospital.cqmu.edu.cn; 3School of Population Medicine and Public Health, Chinese Academy of Medical Sciences & Peking Union Medical College, Beijing 100005, China; jiamengmeng@cams.cn (M.J.); yangweizhong@cams.cn (W.Y.)

**Keywords:** influenza vaccination, health workers, meta-analysis, incidence of laboratory-confirmed influenza, absenteeism rate

## Abstract

A systematic review and meta-analysis was conducted to estimate the pooled effect of influenza vaccinations for health workers (HWs). Nine databases were screened to identify randomized clinical trials and comparative observational studies that reported the effect of influenza vaccination among HWs. The risk ratio (RR), standardized mean difference, and 95% confidence interval (CI) were employed to study the effect size using fixed/random-effect models. Subgroup analyses and sensitivity analyses were conducted accordingly. Publication bias was examined. Sixteen studies (involving 7971 HWs from nine countries) were included after a comprehensive literature search. The combined RR regarding the incidence of laboratory-confirmed influenza was 0.36 (95% CI: 0.25 to 0.54), the incidence of influenza-like illness (ILI) was 0.69 (95% CI: 0.45 to 1.06), the absenteeism rate was 0.63 (95% CI: 0.46 to 0.86), and the integrated standardized mean difference of workdays lost was −0.18 (95% CI: −0.28 to −0.07) days/person. The subgroup analysis indicated that vaccination significantly decreases the incidence of laboratory-confirmed influenza in different countries, study populations, and average-age vaccinated groups. Influenza vaccinations could effectively reduce the incidence of laboratory-confirmed influenza, absenteeism rates, and workdays lost among HWs. It is advisable, therefore, to improve the coverage and increase the influenza vaccination count among HWs, which may benefit both workers and medical institutions.

## 1. Introduction

Seasonal influenza is a public concern globally. It is estimated that there are about one billion cases of influenza worldwide per year, of which 3–5 million are severe cases and 0.29–0.65 million lead to influenza-related respiratory death [1,2]. Health workers (HWs) are at increased risk of influenza virus infection, which may vary depending on occupation or setting [3], and infected HWs could increase the risk of nosocomial infection and community spread [4,5]

Vaccination is the most effective way to prevent influenza [6,7]. The World Health Organization (WHO) considers HWs to be a priority group for influenza vaccination and reinforces this position by supporting countries to develop and implement national, seasonal immunization policies for HWs [7,8]. Most European countries recommend that HWs, at least those involved in direct patient care, are vaccinated against influenza each year [9]. Despite recommendations for influenza vaccination of HWs, vaccine coverage among HWs has remained far below the desired level to achieve herd immunity [10]. Influenza vaccination coverage among HWs has surpassed 75% in the United States (US) [11] and 76.8% in the United Kingdom (UK) [12], but remains below 30% in many other European countries [13], and covered only approximately 5% of HWs in China in 2017–2018 [14,15]. The reasons for low coverage rates among HWs are complex, including concerns about side effects, skepticism about vaccine effectiveness, low awareness of vaccine guidelines, and the misunderstanding that influenza is not serious [16,17]

Evidence of the effectiveness of influenza vaccinations is important to increase the confidence of HWs and provide a rational backdrop for the future development of vaccination policies [18,19]. Several reviews have already evaluated the effectiveness of the influenza vaccine in HWs, with some limitations. For example, in a systematic review, Ng et al. evaluated the effects of seasonal vaccines among HWs based on only three randomized controlled trials (RCTs), and there was insufficient evidence suggesting that receiving an influenza vaccine reduces the incidence of influenza, number of influenza-like illness (ILI) episodes, days with ILI symptoms, or amount of sick leave taken [20]. Another review was a qualitative analysis, without quantitative results [21]. Other reviews evaluated morbidity and mortality outcomes among patients, but did not address the impact for HWs [22,23]. In this study, we performed a systematic review and meta-analysis to comprehensively evaluate and update the evidence on influenza vaccinations for HWs.

## 2. Methods

### 2.1. Study Design

We conducted this systematic review and meta-analysis based on the statement of the Preferred Reporting Items for Systematic and Meta-Analysis (PRISMA) guidelines [24,25]. According to the Centers for Disease Control and Prevention (CDC), we considered physicians, nurses, emergency medical personnel, dental professionals and students, medical and nursing students, laboratory technicians, pharmacists, hospital volunteers, and administrative staff as HWs [26]. The study protocol was registered in the International Prospective Register for Systematic Reviews (PROSPERO, registration number: CRD42020162226) [27].

### 2.2. Search Strategy

The following nine databases were searched for articles meeting the inclusion criteria: PubMed; Web of Science; EMBASE; Scopus; The Cumulative Index to Nursing and Allied Health Literature (CINAHL); The Cochrane Library; China National Knowledge Infrastructure (CNKI); Chinese Science and Technology Periodical Database; WanFang Database; The Scientific Electronic Library Online (SciELO) database. Publications in English, Chinese, and Japanese were eligible for review. This was supplemented by searching ongoing trial databases, such as ClinicalTrials.gov and the WHO International Clinical Trial Search Registry Platform. The reference lists of the included studies and relevant reviews were also manually searched. English- and Chinese-language articles were reviewed by Tingting Li and Xiaoling Qi. This review and meta-analysis was restricted to the published literature. Gray literature was not considered in this study. Gray literature is a field in library and information sciences that deals with the production, distribution, and access to multiple document types produced at all levels of the government, as well as by academics, businesses, and organizations in electronic form; the print formats of gray literature are not controlled by commercial publishing, or publishing is not the primary activity of the producing body [28].

The literature search was performed using combinations of the following search terms: Exp health personnel/OR (health* ADJ2 personnel) OR (health* ADJ2 worker*) OR (health* ADJ2 aid*) OR nurse* OR doctor* OR physician* OR care provider* OR clinician*.

### 2.3. Selection Criteria

Published articles were considered for inclusion if they met the following criteria: (1) The studies were either RCTs or comparative observational studies (cohort and case-control studies). (2) The articles compared the effects of influenza vaccination on HWs with a placebo or non-influenza vaccine control, or unvaccinated HWs. (3) The studies reported one of the following outcomes: incidence of laboratory-confirmed influenza; incidence of ILI; incidence of absenteeism; the number of working days lost per person. (4) The data reported were complete and could be used in the meta-analysis. (5) The data were published in peer-reviewed journals in English, Chinese, or Japanese.

We excluded studies that focused on influenza pandemics. For example, monovalent A (H1N1) pdm09 vaccines that were produced in response to the 2009 global influenza pandemic were excluded. As the focus of this work was specific to HW endpoints, we also excluded studies that solely investigated the effects of vaccines on patient-related outcomes in patients or HWs.

### 2.4. Study Selection and Data Extraction

The literature search was conducted by two teams based on the Cochrane guideline, which were guided by Li Qi and Luzhao Feng, respectively. Identified articles were independently screened by two reviewers (Xiaoling Qi and Tingting Li) for inclusion in the analysis, and for those with titles and abstracts that met the criteria, full-texts were obtained and re-screened. EndNote (X9, Clarivate, Philadelphia, PA, USA) was used to record and manage identified articles. Any discrepancies were resolved during a weekly discussion of reviewed articles by Tingting Li and Xiaoling Qi. In case of disagreement, an independent evaluation was performed by a third experienced reviewer, Li Qi. In total, about 120 published papers were discussed and finally agreed upon.

Data extraction was performed systematically by two reviewers (Tingting Li and Li Qi) using piloted standardized forms, and discrepancies were resolved by discussion to reach a consensus. The extracted data included: published information (first author’s name, published year, and country); study design (RCT or observational study); participants’ characteristics (sample, setting, and profession); intervention information; comparison (placebo, non-influenza vaccine, or unvaccinated HWs); outcomes (incidence of laboratory-confirmed influenza cases or ILI in vaccination groups and comparison groups, absenteeism rate, mean workdays lost, and standard deviation in two groups); other study characteristics (duration of follow-up). Excel (Microsoft Office Professional 2016, Microsoft, Redmond, WA, USA) was utilized to manage the extracted data and information.

### 2.5. Quality Assessment

Two members of our research team (Tingting Li and Xiaoling Qi) independently assessed each included trial, and the risks of bias for RCTs and observational studies were evaluated. For RCTs, the Cochrane Collaboration tool (ROB 2.0) was used to evaluate the quality of studies [29,30,31], by assessing the following five domains: randomization process (including random sequence generation and allocation concealment); deviation from intended interventions; missing outcome data; measurement of the outcome; selection of the reported result. Each RCT was classified into “low risk,” “of some concern,” or “high risk” for each domain, and was determined to be “low risk,” “of some concern,” or “high risk” for overall bias.

For observational studies, the Newcastle-Ottawa Scale (NOS) was employed to evaluate the risk of bias for cohort and case-control studies [32]. The NOS uses the “star system,” with a maximum of nine stars, and evaluates three domains with a total of eight items: selection; comparability; the outcome of exposure of interest. Studies with scores of seven or more stars were considered to be of high quality, five to six stars of moderate quality, and four or fewer stars of low quality [33]

### 2.6. Statistical Analysis

Data analysis was performed with STATA (version 16, StataCorp, College Station TX, USA) and all tests were two-sided. A *P*-value of 0.05 was considered to determine statistical significance unless indicated otherwise. For dichotomous and continuous outcomes, we calculated pooled risk ratios (RRs), or standardized mean differences (SMD) with their 95% confidence intervals (CIs), to examine the effectiveness of influenza vaccination by the fixed-effect model (Mantel–Haenszel method) or the random-effect model (DerSimonian– Laird method) if there was certain heterogeneity. We tested the statistical heterogeneity using Cochran’s Q statistical-based χ^2^ and I-squared (I^2^) statistics [34]. For I^2^ values of 0% to 30%, it was determined that heterogeneity might not be important, while I^2^ values of 30% to 50% indicated moderate heterogeneity, I^2^ values of 50% to 75% indicated substantial heterogeneity, and I^2^ values of 75% to 100% indicated considerable heterogeneity [34,35]. The fixed-effect model (Mantel–Haenszel method) was utilized to calculate the pooled effects when I^2^ < 50%, and the random-effect model (DerSimonian–Laird method) was utilized to calculate the pooled effects when I^2^ ≥ 50% [30]. Considering the low power of Cochran’s Q statistic, a *p*-value of 0.10 was used to determine statistical significance in the interpretation of heterogeneity [30]. Subgroup analyses were performed for the primary outcome and were employed to explore the source of heterogeneity among included studies. Vaccine effectiveness was calculated as 1-RR.

Sensitivity analyses were performed to quantitatively detect the robustness of the results. We estimated the robustness of the pooled effect of influenza vaccination for HWs by excluding studies one by one. Peters’ and Harbord’s tests were applied to determine the publication bias of dichotomous outcomes, including the incidence of laboratory-confirmed influenza, ILI, and absenteeism. Egger’s and Begg’s tests were employed to identify publication bias in continuous outcomes. A *p*-value of 0.05 was considered to be an indicator of statistical significance, which implies that the publication bias among the included studies could not be ignored.

## 3. Results

After initial systematic searches, a total of 10,263 potentially relevant articles were identified. Of these, 2835 citations were removed due to duplication, 6913 reports were excluded based on a review of the title and abstract, 515 citations were fully screened, and 25 reports were included for final identification. Nine studies were excluded because of the lack of availability of the full text (incomplete data extraction, six; inapplicable study design, three). Consequently, a total of 16 studies (19 arms) were included in this systematic review and meta-analysis [36,37,38,39,40,41,42,43,44,45,46,47,48,49,50,51]. Details of the complete selection process of studies are shown in a PRISMA flow chart (Figure 1).

### 3.1. Study Characteristics

A total of 16 studies (19 arms) were included in this study, including six RCTs and ten cohort studies, which were published between 1988–2016. A total of 7971 participants were involved in the 16 studies, including 2745 from vaccinated groups and 5226 from control groups. The studies were carried out in nine countries, of which five were in Japan, two in the US, Italy, and China, separately, and five in other countries. The mean age of the HWs from the vaccinated and comparison groups ranged from 22–45 years old and 22–44 years old, respectively. Table 1 presents the detailed characteristics of the studies.

### 3.2. Quality Assessment

Of the six RCTs, four had moderate risks of bias and two had low risks of bias (Figure 2, Appendix A). Of the ten cohort studies, three displayed high quality and seven exhibited moderate quality (Table 2).

### 3.3. Meta-Analysis

#### 3.3.1. Incidence of Laboratory-Confirmed Influenza

Differences in the incidence of laboratory-confirmed influenza were reported in two RCTs (including five arms) [38,39] and three cohort studies [41,44,47]. A total of 1428 HWs were enrolled, including 898 vaccinated HWs and 530 comparisons. Pooled effects using the fixed-effect model showed that vaccinated HWs were less likely to get an influenza infection than comparisons (pooled RR: 0.36, 95% CI: 0.25 to 0.54, I^2^ = 9.7%; *p* < 0.001, Figure 3), and that vaccine effectiveness (VE) was 64%.

Subgroup analyses were conducted by country, study population, study design, and published year. The results indicated that influenza vaccination could significantly reduce the incidence of laboratory-confirmed influenza in different countries (Appendix A), including the US (RR: 0.13, 95% CI: 0.05 to 0.40), Belgium (RR: 0.48, 95% CI: 0.25 to 0.92), and Japan (RR: 0.54, 95% CI: 0.31 to 0.97). The VE values for the three subgroups were 87%, 52%, and 48%, respectively. Regarding the study population, vaccination could decrease influenza infection among doctors and nurses (RR: 0.13, 95% CI: 0.05 to 0.37, with 87% VE) and doctors alone (RR: 0.48, 95% CI: 0.25 to 0.92, with 52% VE; Appendix A). Moreover, vaccination was effective in reducing the influenza incidence among HWs in studies with different study designs and published years (Appendix A). Additionally, we divided the studies based on the average age of the experimental groups, and the results showed that the influenza vaccination works well for HWs in three age groups. The RRs for HWs aged under 30 years old, 30 to 40 years old, and more than 40 years old were 0.13 (95% CI: 0.04 to 0.48), 0.31 (95% CI: 0.14 to 0.70), and 0.41 (95% CI: 0.22 to 0.79), respectively (Appendix A).

#### 3.3.2. Incidence of IL

Eight studies reported the ILI incidence, including four RCTs [36,43,48,49] and four cohort studies [44,46,50,51], which involved 1543 HWs from vaccination groups and 2211 HWs from comparison groups. The results indicated an overall insignificant reduction of ILI incidence among vaccination groups compared with comparisons (combined RR: 0.69, 95% CI: 0.45 to 1.06, *p* = 0.087, Figure 4). The forest plot showed substantial heterogeneity among these studies (I^2^ = 87.2%). Therefore, we performed subgroup analyses to examine the source of heterogeneity. No statistical significance was found in different research regions: Asia (RR: 0.64, 95% CI: 0.40 to 1.04, I^2^ = 89.1%; *p* = 0.074); study populations: HWs (unspecified; RR: 0.64, 95% CI: 0.37 to 1.08, I^2^ = 84.8%; *p* = 0.093), doctors and nurses (RR: 0.69, 95% CI: 0.45 to 1.06, I^2^ = 87.2%; *p* = 0.789); average ages of vaccinated groups: 30–40 years (RR: 0.63, 95% CI: 0.35 to 1.16, I^2^ = 89.3%; *p* = 0.141), > 40 years (RR: 1.02, 95% CI: 0.38 to 2.75, I^2^ = 77.2%; *p* = 0.967). As for study design, the results demonstrated that a decreased incidence of ILI was reported in RCT studies (RR: 0.52, 95% CI: 0.36 to 0.76, I^2^ = 56.8%; *p* = 0.001, Appendix A). The results also showed that influenza vaccination could effectively reduce the incidence of ILI within half a year of vaccination (RR: 0.45, 95% CI: 0.35 to 0.57, I^2^ = 0.0%; *p* < 0.001, Appendix A).

#### 3.3.3. Absenteeism Rate

Six studies reported discrepancies in the absenteeism rate between vaccinated HWs and comparisons, which included 3475 HWs (824/2651) from the US [36], Australia [37], Italy [40,45], China [42], and Malaysia [43]. Synthesis of the results revealed that the absenteeism rate decreased significantly among HWs exposed to influenza vaccination (overall RR: 0.63, 95% CI: 0.46 to 0.86; *p* = 0.004). However, there was moderate heterogeneity among these studies (I^2^ = 54.3%).

#### 3.3.4. Workdays Lost

The workdays lost among vaccination and comparison groups were reported in five studies [36,38,42,43,46], in which 1500 (686/814) HWs were enrolled. The pooled effect presented a significant decrease in workdays lost for vaccinated HWs in contrast to comparison groups (summarized SMD: −0.18, 95% CI: −0.28 to −0.07, I^2^ = 28.0%; *p* = 0.001).

### 3.4. Sensitivity Analyses and Publication Bias

Sensitivity analyses were performed to determine the robustness of our results. Similar results were observed, indicating the robustness of our results (Appendix A).

The results of Peters’ and Harbord’s tests for the binary outcomes of the incidence of laboratory-confirmed influenza (Peters’: *p* = 0.397; Harbord’s: *p* = 0.082), incidence of ILI (Peters’: *p* = 0. 646; Harbord’s: *p* = 0.339), and absenteeism rate (Peters’: *p* = 0.924; Harbord’s: *p* = 0.167) concluded that publication bias could be ignored among the included studies. The *P*-values of Egger’s test (*p* = 0.682) and Begg’s test (*p* = 0.462) for the continuous outcome of workdays lost per person also indicated that there was no statistically significant difference in publication bias among the included studies.

## 4. Discussion

To the best of our knowledge, this study is the first to provide a comprehensive estimate of the effect of seasonal influenza vaccination on HWs with multiple outcomes, including the incidence of laboratory-confirmed influenza and ILI, absenteeism rate, and mean workdays lost. This study showed the benefits of influenza vaccination for HWs and indicated that vaccination could substantially reduce the incidence of laboratory-confirmed influenza infection by 64%. Furthermore, the results demonstrated that vaccination is also beneficial for medical institutions, and that it may effectively reduce the absenteeism rate among HWs by 37% and lessen the workdays lost by 0.18 days/person among vaccinated HWs in contrast to comparison groups.

In this meta-analysis, we found that the effective reduction of laboratory-confirmed influenza infection among HWs could be observed in different subgroups, including different research districts, study populations, study designs, and published years, with a range of 52–87% VE. Subgroup analyses indicated that vaccination should be implemented in different subgroups, including doctors, nurses, and other HWs. Previous studies indicated that there is some risk of unvaccinated HWs transmitting the virus to patients [52]. As such, vaccination could provide indirect protection to family members and surrounding people [43], which emphasizes the necessity of vaccination for HWs. According to the recommendations of the WHO, it is better to inoculate annually before a local influenza epidemic, and to match the local influenza virus strain [53]

In terms of the countries where studies were conducted, Japan reported the most related studies. Five out of sixteen studies were conducted in Japan, which included two RCTs and three cohort studies. Yet, the studies reported different outcomes: two studies reported the laboratory-confirmed incidence of influenza and three reported the incidence of ILI. The quality of all five studies was moderate, which may affect the accuracy of the results. Therefore, we performed a sensitivity analysis by removing studies conducted in Japan, but the pooled results of incidence of laboratory-confirmed influenza did not change remarkably (pooled RR: 0.27, 95% CI: 0.16 to 0.47; I^2^ = 0.5%, *p* < 0.001), and the combined results of ILI incidence also showed only an insignificant change (integrated RR: 0.92, 95% CI: 0.61 to 1.39; I^2^ = 82.2%, *p* = 0.685).

Our study suggests that ILI is not a recommended single indicator for evaluating the effectiveness of an influenza vaccination, and that a combination of laboratory data is needed. First, the definition of ILI is not unified. The CDC defines ILI as “a temperature of ≥100.0 °F (≥37.8 °C), oral or equivalent, and cough or sore throat, in the absence of a known cause other than influenza” [36,54], but there are other definitions in our included studies, which define ILI as “a febrile illness involving fever for a minimum of 1 day plus two or more of the following symptoms: cough, sore throat, runny nose, myalgia, malaise or headache” [43], or “acute onset of high fever (axillary temperature ≥38 °C), accompanied by 2 of the following symptoms, such as chills, weakness, headache, myalgia or joint pain, cough, sore throat and nasal congestion, nasopharynx redness and swelling” [50]. Besides, other respiratory viruses could also present as ILI, which makes the clinical differentiation of influenza from other pathogens difficult [55]. In addition, the assessment of ILI depends on the clinical reporting of patients, which may cause bias if inaccurate data are provided [56]. Finally, a previous study concluded that ILI has no role in measuring influenza VE [57]. This may explain the substantial heterogeneities in the pooled results of the incidence of ILI.

Our findings provide convincing evidence of the effect of the influenza vaccination for HWs. Vaccination is imperative, especially for countries and areas in which influenza vaccination coverage rates are low for HWs, like China [14,15] and Italy [13]. Besides the reports included in this study, other excluded citations with incomplete data also reported that influenza vaccination was considerably useful in limiting the length of absenteeism [58,59,60] and workdays lost [58,59,60,61]. Furthermore, vaccination is cost-efficient for medical institutions since it may save overall costs for hospitals or other medical departments, given that expenditure on absent HWs caused by influenza totals more than the cost of vaccination activities [40,50,59,60,62,63], which is in accordance with a previous meta-analysis [33].

Some actions might be conducive to improving influenza vaccine coverage among HWs. First, national policies and normative guidelines regarding influenza vaccination for HWs should be published as soon as possible, as studies indicate that mandatory influenza vaccination policies are more effective than nonmandatory policies in reducing the rate of absenteeism among HWs [64,65]. Second, financial subsidies for influenza vaccination may ensure that HWs can be inoculated for free. Alternatively, it is advisable that the medical insurance of HWs should fully, or mostly, cover the costs of influenza vaccinations. One study indicated that some local reimbursement policies regarding influenza vaccination only covered a small part of the population in China [66]. Third, the investment in scientific research and development for the influenza vaccine should be increased, to improve the efficacy of the vaccine, since some reports indicated that the VE was low in many countries, like Japan [47] and Canada [67]. Fourth, the awareness of HWs regarding influenza vaccinations should be improved, as awareness promotes a positive attitude that results in improved practices. Further effort is necessary to increase the HWs’ awareness regarding influenza vaccination. Suitable awareness initiatives include, for instance, sending text-message/e-amils when influenza season comes, implementing peer support (i.e., support from others HWs with high awareness of influenza vaccinations), conducting targeted training courses to pursue the academic detailing methodology [68,69], providing in-service education [70,71], and implementing awareness campaigns [68,72]. Last, vaccination healthcare services should be improved, such as vaccine supply, accessibility of services, and notifications of potential side effects.

There are several limitations to our study. First, only sixteen studies from nine countries were included in the meta-analysis, and papers published in other languages were not taken into consideration, so the results cannot be generalized without further validation. Moreover, sixteen studies were conducted before the outbreak of COVID-19. Some reports have indicated that COVID-19 and influenza have coinfections [73,74,75], which might affect seasonal influenza vaccination effects. Second, only three high-quality studies were included. More high-quality studies are required to explore influenza vaccination effectiveness among HWs. Third, for the secondary outcomes of the absenteeism rate and workdays lost, the included studies and sample sizes were relatively small, and there was heterogeneity of the work structure between healthcare settings, which might affect the interpretation of our results regarding the two outcomes. Overall, more studies are needed to provide scientific evidence in more countries.

## 5. Conclusions

This systematic review and meta-analysis provides evidence that influenza vaccination plays a crucial role in reducing the incidence of laboratory-confirmed influenza, absenteeism rates, and mean workdays lost among HWs. It is advisable to improve the coverage and increase influenza vaccination counts for HWs, which may benefit HWs and medical institutions alike.

## Figures and Tables

**Figure 1 vaccines-09-01104-f001:**
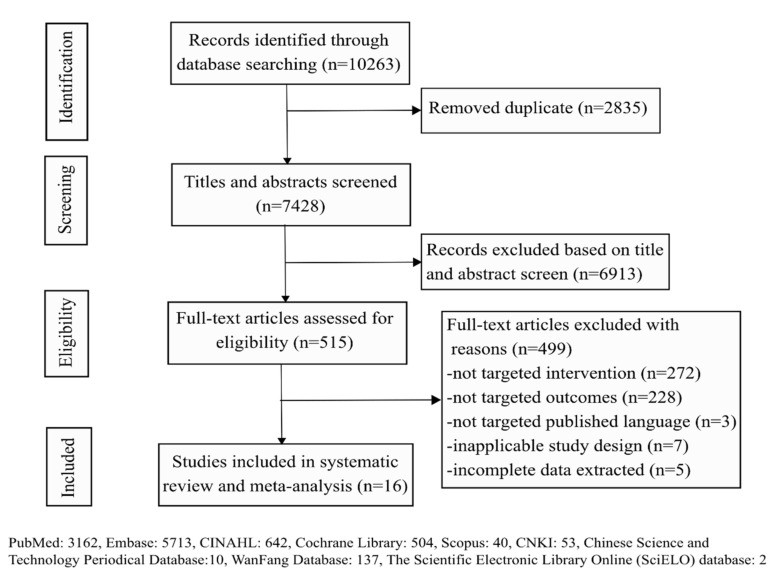
Flow diagram of study selection.

**Figure 2 vaccines-09-01104-f002:**
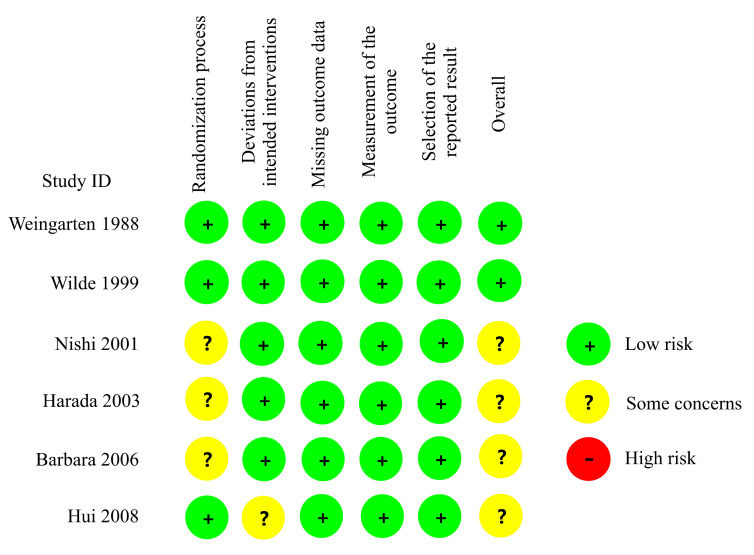
Quality assessment of the six randomized controlled trials. ID, identification.

**Figure 3 vaccines-09-01104-f003:**
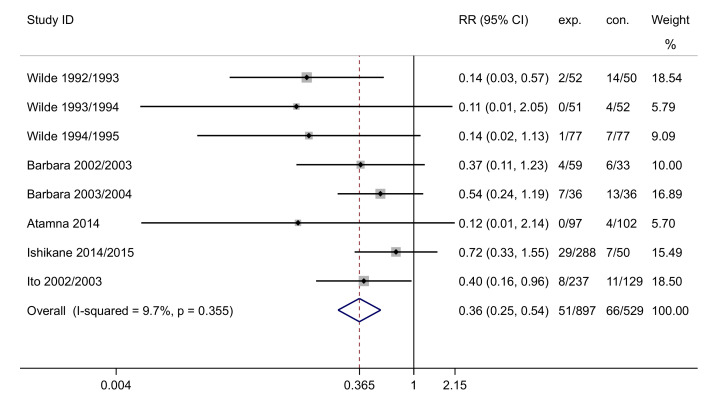
Forest plot of incidence of laboratory-confirmed influenza. ID, identification; exp., experimental group; con., control group; RR, risk ratio.

**Figure 4 vaccines-09-01104-f004:**
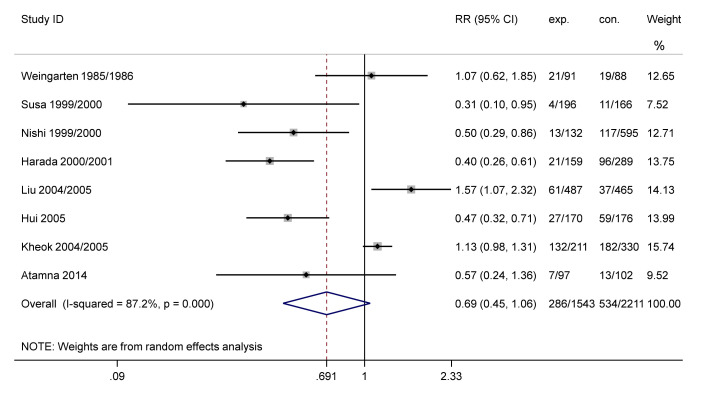
Forest plot of ILI incidence. ILI, influenza-like illness; ID, identification; exp., experimental group; con., control group; RR, risk ratio.

**Table 1 vaccines-09-01104-t001:** Detailed information of included studies in the meta-analysis.

Author, Published Year	Conducted Year	Published Language	Country	Source	Study Population	Age Mean (SD) exp./con.	Sample Size (exp./con.)	Comparison	Study Design	Follow-Up Period
Weingarten, 1988 [36]	1985/1986	English	the US	Hospital	HWs ^a^	35.8 (8.9)/37.0 (9.0)	179 (91/88)	Placebo (saline)	RCT	14 m
Jones, 1999 [37]	1998	English	Australia	Hospital	HWs ^b^	-	93 (46/47)	Unvaccinated	cohort study	5 m
Wilde, 1999 [38]	1992/1993	English	the US	Hospital	HWs ^b^	28.4 (2.6)/28.9 (2.8)	102 (52/50)	Meningococcal vaccine	RCT	6 m
Wilde, 1999 [38]	1993/1994	English	the US	Hospital	HWs ^b^	28.0 (2.8)/28.3 (3.3)	103 (51/52)	Pneumococcal vaccine	RCT	6 m
Wilde, 1999 [38]	1994/1995	English	the US	Hospital	HWs ^b^	30.0 (5.1)/30.8 (5.9)	156 (78/78)	Placebo (saline)	RCT	6 m
Susa, 2001 [51]	1999/2000	Japanese	Japan	Hospital	HWs ^a^	38.2 (10.8)/41.2 (6.2), 40.8 (12.0)	362 (196/166)	Unvaccinated	cohort study	4 m
Nishi, 2001 [48]	1999/2000	Japanese	Japan	Hospital	HWs ^a^	39.5 (9.2)/37.1 (10.2)	727 (132/595)	Unvaccinated	RCT	3 m
Harada, 2003 [49]	2000/2001	Japanese	Japan	Hospital	HWs ^a^	36.3/38.1	348 (159/189)	Unvaccinated city office staff	RCT	4 m
Barbara, 2006 [39]	2002/2003	English	Belgium	Hospital	HWs ^c^	42.4 (1.1)/40.0 (1.5)	92 (59/33)	Unvaccinated	RCT	6 m
Barbara, 2006 [39]	2003/2004	English	Belgium	Hospital	HWs ^c^	43.1 (0.9)/41.3 (10.8)	72 (36/36)	Unvaccinated	RCT	5 m
Colombo, 2006 [40]	2002/2003	English	Italy	Healthcare unit	HWs ^d^	44.3/43.2	214 (107/107)	Unvaccinated	cohort study	4.5 m
Ito, 2006 [41]	2002/2003	English	Japan	Hospital	HWs ^a^	35.5 (10.3)/35.7 (9.4)	366 (237/129)	Unvaccinated	cohort study	4 m
Liu, 2006 [50]	2004/2005	Chinese	China	Hospital	HWs ^a^	40.5 (10.3)/38.4 (10.9)	952 (487/465)	Unvaccinated	cohort study	6 m
Chan, 2007 [42]	2004/2005	English	China	Emergency department	HWs ^e^	43.55 (8.85)/40.65 (6.99)	73 (33/40)	Unvaccinated	cohort study	10 m
Hui, 2008 [43]	2005	English	Malaysia	Faculty of Dentistry	HWs ^a^	22.0 (3.52)/22.5 (4.90)	346 (170/176)	Unvaccinated	RCT	4 m
Kheok, 2008 [46]	2004/2005	English	Singapore	Hospital	HWs ^b^	37 (11.3)/33 (9.1)	541 (211/330)	Unvaccinated	cohort study	12 m
Amodio, 2010 [45]	2007/2008	English	Italy	Hospital	HWs ^d^	49.9 (8.6)/47.2 (9.3)	2608 (215/2393)	Unvaccinated	cohort study	3 m
Atamna, 2016 [44]	2014	English	Israel	Hospital	HWs ^b^	43.15 (12.06)/39.32 (10.85)	199 (97/102)	Unvaccinated	cohort study	4 m
Ishikane, 2016 [47]	2014/2015	English	Japan	Long-term care facility	HWs ^e^	-	338 (288/50)	Unvaccinated	cohort study	1 m

Note: the US, the United States; RCT, randomized controlled trial; SD, standard deviation; exp., experimental group; con., control group; m, month(s); HWs ^a^, HWs (unspecified); HWs ^b^, doctors and nurses; HWs ^c^, doctors; HWs ^d^, doctors, nurses, and administrators; HWs ^e^, nurses.

**Table 2 vaccines-09-01104-t002:** Quality assessment of ten cohort studies.

Author, Published Year	Selection				Comparability	Outcome			Total
	Item 1	Item 2	Item 3	Item 4	Item 5	Item 6	Item 7	Item 8	
Jones, 1999	**✵**	**✵**	**✵**	**✵**	×	**✵**	×	**✵**	6 stars
Colombo, 2006	**✵**	**✵**	**✵**	**✵**	**✵** **✵**	**✵**	×	**✵**	8 stars
Ito, 2006	**✵**	**✵**	**✵**	**✵**	×	×	×	**✵**	5 stars
Liu, 2006	**✵**	**✵**	**✵**	**✵**	**✵** **✵**	×	**✵**	**✵**	8 stars
Susa, 2006	**✵**	**✵**	**✵**	**✵**	×	×	×	**✵**	5 stars
Chan, 2007	**✵**	**✵**	**✵**	**✵**	×	**✵**	×	**✵**	6 stars
Kheok, 2008	**✵**	**✵**	**✵**	**✵**	×	**✵**	**✵**	**✵**	7 stars
Amodio, 2010	**✵**	**✵**	**✵**	**✵**	×	**✵**	×	**✵**	6 stars
Atamna, 2016	**✵**	**✵**	**✵**	**✵**	×	**✵**	×	**✵**	6 stars
Ishikane, 2016	**✵**	**✵**	**✵**	**✵**	×	**✵**	×	**✵**	6 stars

Note: Item 1, Representativeness of the exposed cohort; Item 2, Selection of the non-exposed cohort; Item 3, Ascertainment of exposure; Item 4, Demonstration that outcome of interest was not present at start of study; Item 5, Comparability of cohorts on the basis of the design or analysis; Item 6, Assessment of outcome; Item 7, Whether follow-up was long enough for outcomes to occur; Item 8, Adequacy of follow-up of cohorts. **✵**, the included study met the criteria of this item; ×, the included study did not meet the criteria of this item.

## Data Availability

Data sharing is not applicable in this systematic review and meta-analysis. Data used in this study are available from the included published papers.

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
