# Peer review of "A Systematic Review and Meta-Analysis of Seasonal Influenza Vaccination of Health Workers"

_vaccines, 2021, doi:10.3390/vaccines9101104_

Round 1

Reviewer 1 Report

The manuscript "The effect of seasonal influenza vaccination on health workers: a systematic review and meta-analysis" is a systematic review and meta-analysis that examines the impact of influenza vaccination on a specific risk group.

As clearly specified by the authors, this is the first study that aims at providing a comprehensive estimate of the effect of seasonal influenza vaccination on HWs: this is a topic of crucial importance, especially in times when - due to the pandemic and the controversy regarding the compulsory vaccinations of healthcare workers - the discussion about the importance of protecting them is raging.

It is a well-written, organized and significant work. 

The readability is good and the overall quality is high. 

The authors were able to discuss the topic in depth, enclosing an excellent reference list. Methods are well described. A complete analysis was carried out without losing the sight on the main focus. 

Some minor comments:

  • The groups of participants included in the different studies have different ages, as shown in the table. I would like to ask the authors if they found or noticed any differences in vaccination effectivness in groups of vaccinated with different mean ages.
  • In the “discussion” section, the authors list a number of important actions to consider to improve influenza vaccination coverage among HWs. Regarding the fourth (“the awareness of HWs regarding influenza vaccinations should be improved, as awareness promotes a positive attitude that results in improved practices”), I would suggest to better detail, based on current scientific evidence, what actions could be taken to achieve this objective. I think Vaccines readers would certainly be interested.
  • For a better understanding of the included groups, I would suggest to add a column with “types of workers included in the study” (or similar) to table 1, reporting if the study included doctors or nurses or both (or students, dentists, etc.).

The readability is good and the overall quality is high. 

The authors were able to discuss the topic in depth, enclosing an excellent reference list. Methods are well described. A complete analysis was carried out without losing the sight on the main focus. 

My recommendation is to accept this paper from Li T. and colleagues. 

Author Response

Responses to Reviewer 1:

  1. The groups of participants included in the different studies have different ages, as shown in the table. I would like to ask the authors if they found or noticed any differences in vaccination effectiveness in groups of vaccinated with different mean ages.

Response: We appreciate this valuable comment. As per the suggestion, we conducted a subgroup analysis based on the average age of HWs regarding the two primary outcomes. The results showed the influenza vaccination works well for HWs in different age groups in reducing the incidence of lab-confirmed influenza, but not consistent in different age groups regarding the incidence of ILI. More details were provided in the results section and supplement files. (Page 9, paragraph 2 and 3; Fig. S2e)

  1. In the “discussion” section, the authors list a number of important actions to consider to improve influenza vaccination coverage among HWs. Regarding the fourth (“the awareness of HWs regarding influenza vaccinations should be improved, as awareness promotes a positive attitude that results in improved practices”), I would suggest to better detail, based on current scientific evidence, what actions could be taken to achieve this objective. I think Vaccines readers would certainly be interested.

Response: Thanks for pointing out this. We added some suggestions aiming to achieve this objective based on current scientific evidence as follows. (Page 12, paragraph 1)

Further effort is necessary to increase the HWs’ awareness regarding influenza vaccination. For instance, send text-message/e-mails when influenza season comes, implement peer support which means support from others HWs with high awareness of influenza vaccinations, conduct targeted training courses by academic detailing methodology,68,69 provide in-service education,70,71 and implement awareness campaigns.68,72

Reference list:

  1. Bert F, Thomas R, Lo Moro G, et al. A new strategy to promote flu vaccination among health care workers: Molinette Hospital's experience. J Eval Clin Pract. 2020;26(4):1205-1211.
  2. Tamburrano A, Mellucci C, Galletti C, et al. Improving Nursing Staff Attitudes toward Vaccinations through Academic Detailing: The HProImmune Questionnaire as a Tool for Medical Management. Int J Environ Res Public Health. 2019;16(11).
  3. Alshammari TM, Yusuff KB, Aziz MM, Subaie GM. Healthcare professionals' knowledge, attitude and acceptance of influenza vaccination in Saudi Arabia: a multicenter cross-sectional study. BMC Health Serv Res. 2019;19(1):229.
  4. Andayi F, Emukule GO, Osoro E, et al. Knowledge and attitude of Kenyan healthcare workers towards pandemic influenza disease and vaccination: 9 years after the last influenza pandemic. Vaccine. 2021;39(29):3991-3996.
  5. Choucair K, El Sawda J, Assaad S, et al. Knowledge, Perception, Attitudes and Behavior on Influenza Immunization and the Determinants of Vaccination. J Epidemiol Glob Health. 2021;11(1):34-41.

  1. For a better understanding of the included groups, I would suggest to add a column with “types of workers included in the study” (or similar) to table 1, reporting if the study included doctors or nurses or both (or students, dentists, etc.).

Response: Good point. As per your suggestion, we provided more details of the types of workers in Table 1. There are five types of HWs: HWs a: Doctors, nurses, and administrators; HWs b: Doctors and nurses; HWs c: Doctors; HWs d: Nurses; HWs e: HWs (unspecified). Please see Table 1 in the enclosed revised manuscripts for details. (Page 6)

Reviewer 2 Report

Overview:

This manuscript depicts the methods and the results of a review and meta-analysis of the published literature (English, Japanese, Chinese) on vaccination of healthcare workers. One of the novel aspects of this research is that it sought to evaluate work-related consequences of health workers becoming infected with influenza.

The authors provide significant detail of the methods used to search and refine the analysis. The authors use several statistical tools to evaluate the studies that met all their criteria for inclusion in the meta-analysis. The methods and the statistical evaluation of the data seem appropriate given the scope of this research project.

The research is a significant contribution to the literature as a result of the breadth of the analysis, especially as it relates to the inclusion of Japanese and Chinese scientific literature. The observation influenza-like-illnesses alone are not statistically relevant indicators for evaluation of the effectiveness of influenza vaccines is a relatively novel insight in this type of study.

The authors conclusion that increasing immunization rates for health workers, especially, in non-US and non-European countries, is worthy of pointed interest. The results of this research perhaps will provide a rationale for increases in annual influenza vaccinations in countries such as Italy and China (as detailed by the authors) as well as others. However, the barriers to vaccination among health workers are covered by others and addressing such issues is beyond the scope of the current research.

Overall, this is a thorough paper. It is well-organized and well-written. 

Title:

Perhaps the title could be changed to ‘A systematic review and meta-analysis of seasonal influenza vaccination on health workers.’ This would reflect the fact that the timing of immunization was not a factor in the analysis performed. Instead, the authors focused their review on the clinical outcomes (including work records) of annually vaccinated and unvaccinated health workers.

Abstract:

The first two sentences perhaps could be eliminated. However, in defense of the current manuscript, the current format does mimic other review articles on this topic.

Introduction:

The authors do a good job of providing a rationale for the current research.

Materials, Methods, & Results:

  • Section 2.4: what was the between scorer concordance between Xiaoling Qi and Li? In other words, how often was there disagreement as to what published papers should be included in the meta-analysis?
  • Section 2.4: Did Li Qi evaluate articles independently to settle disagreements or did Li Qi decide ties based on the arguments presented by the other authors? How often did such an arbitration take place?
  • Is there a reason no studies matched the inclusion criteria from times earlier than 1988 or after 2016? What was going on in this 25-30 year period that was different than earlier or afterwards?
  • Figures 3, 4, & 5 may be unnecessarily complicated. The presentation distracts from the otherwise straight forward presentation of the results of the meta-analysis. The authors should consider the importance of keeping these figures. Perhaps there is a way to simplify the figures so that only the main points are depicted. Less important aspects of the figures perhaps could be moved to the body of the text.

Discussion:

Interpretation of the absenteeism rates and workdays lost is complicated by the heterogeneity of work structure between healthcare settings. That is, not all health workers have the same schedule or are susceptible to the same consequences of taking time off from work. These limitations make interpretation of the 0.18 days/person value from the analysis subject to potential criticism. Perhaps the value calculated can be de-emphasized without losing the main point?

The authors may want to consider the appropriateness of the second to last paragraph of the Discussion. It may be beyond the scope of the current research to tell local/national governments, or suggest to local/national governments, how to interpret and implement the results of the current meta-analysis. Is it better to let the data stand on its’ own?

In the last paragraph, please define/explain what the authors mean by ‘gray literature’. Also, this paragraph concludes by questioning the ‘accuracy’ of the results. Surely, the analysis performed is accurate based on the data obtained. A better word for the points the authors are making might be ‘generalizability’ of the results. The results of this research would certainly be more generalizable across the world if more studies were available and more countries were represented in the dataset.

Conclusion:

No specific comments.

Author Response

Responses to Reviewer 2:

  1. Title: Perhaps the title could be changed to ‘A systematic review and meta-analysis of seasonal influenza vaccination on health workers.’ This would reflect the fact that the timing of immunization was not a factor in the analysis performed. Instead, the authors focused their review on the clinical outcomes (including work records) of annually vaccinated and unvaccinated health workers.

Response: We appreciate your suggestion and revised the title accordingly.

  1. Abstract: The first two sentences perhaps could be eliminated. However, in defense of the current manuscript, the current format does mimic other review articles on this topic.

Response: Thanks for pointing out this. We revised the abstract in order to make it more concise. (Page 1, Abstract section)

  1. Section 2.4: what was the between scorer concordance between Xiaoling Qi and Li? In other words, how often was there disagreement as to what published papers should be included in the meta-analysis?

Response: Tingting Li and Xiaoling Qi discussed the reviewed articles once per week. In total, about 120 published papers were discussed to make sure whether they should be included or not. We added this sentence in our revised manuscript. (Page 3, paragraph 5)

  1. Section 2.4: Did Li Qi evaluate articles independently to settle disagreements or did Li Qi decide ties based on the arguments presented by the other authors? How often did such an arbitration take place?

Response: Thank you for this kind consideration. Li Qi evaluated articles independently and made such arbitration about 50 times.

  1. Is there a reason no studies matched the inclusion criteria from times earlier than 1988 or after 2016? What was going on in this 25-30 year period that was different than earlier or afterwards?

Response: Some reasons might explain this phenomenon, including the lack of influenza vaccine in many countries before 1988 especially in developing countries, let alone related scientific research. We searched several related studies after 2016, which didn’t meet our inclusion criteria. For instance, Zaffina et al. conducted a retrospective observational study to examine absenteeism in health-care workers in two subsequent influenza seasons, and Martín-Rodríguez et al. implemented a cross-sectional study to estimate the effect of the influenza vaccination in primary healthcare workers, however both of them did not meet our inclusion criteria.

Reference list:

Zaffina S, Gilardi F, Rizzo C, et al. Seasonal influenza vaccination and absenteeism in health-care workers in two subsequent influenza seasons (2016/17 and 2017/18) in an Italian pediatric hospital. Expert Rev Vaccines. 2019; 18(4): 411–418. doi: 10.1080/14760584.2019.1586541.

Martín-Rodríguez MDM, Díaz-Berenguer JA, Alonso-Bilbao JL, et al. Effect of influenza vaccination in Primary Healthcare workers and the general population in Gran Canaria: A cross-sectional study. Aten Primaria. 2019; 51(5): 300–309. Spanish. doi: 10.1016/j.aprim.2017.11.010.

  1. Figures 3, 4, & 5 may be unnecessarily complicated. The presentation distracts from the otherwise straight forward presentation of the results of the meta-analysis. The authors should consider the importance of keeping these figures. Perhaps there is a way to simplify the figures so that only the main points are depicted. Less important aspects of the figures perhaps could be moved to the body of the text.

Response: We appreciate your suggestion. The most important outcomes were incidence of lab-confirmed influenza and incidence of ILI, which were presented in Figure 3, thus we deleted two forest plots of secondary outcomes in original Figure 3. Also, we revised Figure 4 and Figure 5 accordingly and provided the less important aspects of the figures in the supplementary files (Fig. S4a-e and Fig. S5a, b). All forest plots in supplementary files were showed separately instead of combination version. (Page 9 and 10)

  1. Discussion: Interpretation of the absenteeism rates and workdays lost is complicated by the heterogeneity of work structure between healthcare settings. That is, not all health workers have the same schedule or are susceptible to the same consequences of taking time off from work. These limitations make interpretation of the 0.18 days/person value from the analysis subject to potential criticism. Perhaps the value calculated can be de-emphasized without losing the main point?

Response: Thanks. We aimed to comprehensively evaluate the effectiveness of influenza vaccination in this study. Therefore, the combined absenteeism rates and workdays lost were provided even though there was heterogeneity of work structure between healthcare settings. We added this limitation in the revised manuscript. (Page 12, paragraph 2)

  1. Discussion: The authors may want to consider the appropriateness of the second to last paragraph of the Discussion. It may be beyond the scope of the current research to tell local/national governments, or suggest to local/national governments, how to interpret and implement the results of the current meta-analysis. Is it better to let the data stand on its’ own?

Response: The point is well taken. We revised the discussion section thoroughly which might be better. (Page 12, paragraph 1)

  1. Discussion: In the last paragraph, please define/explain what the authors mean by ‘gray literature’. Also, this paragraph concludes by questioning the ‘accuracy’ of the results. Surely, the analysis performed is accurate based on the data obtained. A better word for the points the authors are making might be ‘generalizability’ of the results. The results of this research would certainly be more generalizable across the world if more studies were available and more countries were represented in the dataset.

Response: Many thanks for this valuable comment. As per your suggestion, we revised the discussion section to further clarify this point. (Page 12, paragraph 2)

In addition, please find the definition of gray literature in the Method section in the revised manuscript. (Page 3, paragraph 1)

Reviewer 3 Report

Thank you for giving me the opportunity to review the article. The authors conducted a study focused on the effect of seasonal influenza vaccination on health workers. The topic is socially and clinically important, but several problems existed in tis manuscript. Therefore, the reviewer though that the authors should revised the manuscript before further considerations. I listed the comments below.

Comments:

  1. The authors should check the way to write the e-mail addresses, and correct before submitting the revised manuscript.

Materials and Methods:

  1. Did the authors also conduct the search process by two independent researchers? The reviewer understood that they performed the review process by two researchers, but cannot know about the search process.
  2. The authors should do the hand search process to cover the articles which only mentioned in the references of the extracted articles.
  3. The authors should clearly state the definition of the HWs.
  4. The authors should add the details of the models they used in this study (“fixed” and “mixed” only mean the types of the statistical model).
  5. The authors should write the details of sensitivity analyses in the Methods section.

Results:

  1. The authors should clearly state the number of the records in each database in the Figure 1.
  2. Why the number of the article with irrelative theme was so large? Why were these articles not excluded in the screening process?

Discussion:

  1. Five out of fifteen studies were conducted in Japan. It can affect the results, and the authors should discuss about it.
  2. The authors mainly focused on the studies in hospital setting. The healthcare systems are different among countries, and can it affect the results?
  3. The seasonal influenza infection can be changed in the with-Covid era. Therefore, the authors should mention about it in this section.

Author Response

Responses to Reviewer 3:

  1. The authors should check the way to write the e-mail addresses, and correct before submitting the revised manuscript.

Response: Thank you for pointing out this. As per your suggestion, we revised the e-mail addresses of all co-authors accordingly.

  1. Did the authors also conduct the search process by two independent researchers? The reviewer understood that they performed the review process by two researchers, but cannot know about the search process.

Response: Thanks for your comments. The literature search was conducted by two teams based on the guideline of Cochrane, which were guided by Li Qi and Luzhao Feng respectively. (Page 3, paragraph 5)

  1. The authors should do the hand search process to cover the articles which only mentioned in the references of the extracted articles.

Response: Thanks for pointing out this. As you mentioned that the hand search process is very important. The reference lists of included studies and relevant reviews were hand-searched in our review. The sentence in the 2.2 Search strategy of Method section “The reference lists of included studies and relevant reviews were also searched.” has been changed into “The reference lists of the included studies and relevant reviews were also manually searched.” (Page 2, paragraph 5)

  1. The authors should clearly state the definition of the HWs.

Response: As per your suggestion, we provided the definition of HWs as well as the reference in the revised manuscript. (Page 2, paragraph 4)

According to the Centers for Disease Control and Prevention (CDC), we considered physicians, nurses, emergency medical personnel, dental professionals and students, medical and nursing students, laboratory technicians, pharmacists, hospital volunteers, and administrative staff as HWs.26

Reference list:

  1. Centers for Disease Control and Prevention. https://www.cdc.gov/vaccines/adults/rec-vac/hcw.html

  1. The authors should add the details of the models they used in this study (“fixed” and “mixed” only mean the types of the statistical model).

Response: Thanks for point out this. More details of the models were provided in the method section in the revised manuscript accordingly. (Page 4, paragraph 2)

  1. The authors should write the details of sensitivity analyses in the Methods section.

Response: As per your suggestion, we revised the Methods section to further clarify the sensitivity analysis. (Page 4, paragraph 3)

  1. The authors should clearly state the number of the records in each database in the Figure 1.

Response: Thanks for pointing out this. We revised the Figure 1 accordingly. (Page 5)

  1. Why the number of the article with irrelative theme was so large? Why were these articles not excluded in the screening process?

Response: Thanks for point out this. It is our carelessness in drafting the original manuscript, and we did not find this typo in reviewing and revising process. We corrected the expression in the revised Figure 1. (Page 5)

  1. Five out of fifteen studies were conducted in Japan. It can affect the results, and the authors should discuss about it.

Response: This point is well taken. We conducted a sensitivity analysis by deleting studies conducted in Japan, the pooled results of incidence of lab-confirmed influenza did not change reversely (pooled RR: 0.27, 95% CI: 0.16 to 0.47; I2 = 0.5%, p < 0.001). Moreover, the combined results of incidence of ILI (integrated RR: 0.92, 95% CI: 0.61 to 1.39; I2 = 82.2%, p = 0.685) showed similar after removing studies conducted in Japan. We added this discussion in the revised manuscript. (Page 11, paragraph 3)

  1. The authors mainly focused on the studies in hospital setting. The healthcare systems are different among countries, and can it affect the results?

Response: Thanks for pointing out this. As you mentioned that the healthcare systems are different among countries, so we conducted a subgroup analysis by conducted country. The results showed that influenza vaccination could significantly reduce the incidence of lab-confirmed influenza in different countries, including The US (RR: 0.13, 95% CI: 0.05 to 0.40), Belgium (RR: 0.48, 95% CI: 0.25 to 0.92), and Japan (RR: 0.54, 95% CI: 0.31 to 0.97). (Page 9, paragraph 2; Fig. S2a)

  1. The seasonal influenza infection can be changed in the with-Covid era. Therefore, the authors should mention about it in this section.

Response: This point is well taken. As you mentioned, the seasonal influenza infection can be changed in the with-Covid era, therefore, it is necessary to conduct more studies after the Covid-19 outbreak. We revised the discussion section accordingly. (Page 12, paragraph 2)

Round 2

Reviewer 3 Report

Thank you for giving me the opportunity to review the article. The authors revised the manuscript according to the comments. Therefore, the reviewer though that the manuscript can be accepted for publication.